# The Influence of Short Coir, Glass and Carbon Fibers on the Properties of Composites with Geopolymer Matrix

**DOI:** 10.3390/ma14164599

**Published:** 2021-08-16

**Authors:** Kinga Korniejenko, Michał Łach, Janusz Mikuła

**Affiliations:** Chair of Materials Engineering, Faculty of Material Engineering and Physics, Cracow University of Technology, Jana Pawła II 37, 31-864 Cracow, Poland; michal.lach@pk.edu.pl (M.Ł.); janusz.mikula@pk.edu.pl (J.M.)

**Keywords:** geopolymer, fibers, mechanical properties

## Abstract

The aim of the article is to analyze the influence of short coir, glass and carbon fiber admixture on the mechanical properties of fly ash-based geopolymer, such as: flexural and compressive strength. Glass fiber and carbon fibers have been chosen due to their high mechanical properties. Natural fibers have been chosen because of their mechanical properties as well as for the sake of comparison between their properties and the properties of the artificial ones. Fourth series of fly ash-based geopolymers for each fiber was cast: 1, 2, and 5% by weight of fly ash and one control series without any fibers. Each series of samples were tested on flexural and compressive strength after 7, 14, and 28 days. Additionally, microstructural analysis was carried out after 28 days. The results have shown an increase in compressive strength for composites with fibers—an improvement in properties between 25.0% and 56.5% depending on the type and amount of fiber added. For bending strength, a clear increase in the strength value is visible for composites with 1 and 2% carbon fibers (62.4% and 115.6%). A slight increase in flexural strength also occurred for 1% addition of glass fiber (4.5%) and 2% addition of coconut fibers (5.4%). For the 2% addition of glass fibers, the flexural strength value did not change compared to the value obtained for the matrix material. For the remaining fiber additions, i.e., 5% glass fiber as well as 1 and 5% coconut fibers, the flexural strength values deteriorated. The results of the research are discussed in a comparative context and the properties of the obtained composites are juxtaposed with the properties of the standard materials used in the construction industry.

## 1. Introduction

Nowadays, geopolymer composites are noticed as an environmentally friendly alternative to traditional building materials [1,2]. The geopolymers have good mechanical properties, including compressive strength, resistance to corrosive environments, and high temperatures [2,3]. Because of that they find the application in many areas, such as: fire proof products, absorbents, insolation materials and others (Table 1).

The main weakness of these types of composites is brittle cracking, which limits their use in many areas [2,4]. The main possibility to avoid this phenomenon is the incorporation of fibers as reinforcement into a geopolymer matrix [2,5]. The fibers change the character of the fracture from brittle to more ductile [2,6]. Moreover, they increase a fracture toughness, flexural strength [6,7] and rise an amount of energy absorbed by a material before damage, including a reduction of cracks (size and numbers) [8,9]. The implementation of fibers into the geopolymer matrix creates a very promising material for the construction industry. It is not only because of the outstanding composite properties, but also because of the simple application in the manufacturing process. The short fibers, due to their easy fiber dispersion, are an effective way to strengthen geopolymers, without additional equipment for the production process. [10,11]. The designed composites also help with finding new, more advanced application or help with utilization fiber waste from other industries [12,13,14].

The main aim of the article is to analyze the possibilities of reinforcement of geopolymer composites by different kinds of fibers, including waste fibers such as coir. It is usually waste product from the coconut industry. The composites have a form of paste, without aggregates. Three types of fibers with different properties were selected for the study, including natural fiber derived from coconut and chemical—inorganic fibers: glass fiber and carbon fiber. The selection of fibers for the tests was based on their properties and on the basis of a literature review. The glass and carbon fibers were previously investigated as a reinforcement for the geopolymer composites, in the case of the coir fibers, only a few research were made. They were not so detailed as presented in the article. Next, the properties of the composites are determinate, the opportunities of using a new composite in practical applications, especially in the building industry are presented.

## 2. Materials and Methods

### 2.1. Materials

The raw material for the production of the geopolymer matrix was fly ash supplied from the CEZ Skawina heat and power plant (Poland). The material was in the form of fine, dark gray mineral dust. The analysis of the oxide composition (Table 2) allows for the classification of this ash as fly ash of class F. Elemental composition confirms the possibility of its use as a material for the geopolymerization process. In this aspect, the low amount of calcium in the fly ash and the high amount of aluminum are particularly important [15].

Physical properties and particle morphology are also important factors influencing the properties of the material. The supplied ash has favorable physical properties for the formation of geopolymeric materials, including mostly a fine particle fraction (Table 3), a fineness of 16.7%, a density of 2.22 g/cm^3^ and an index of pozzulan activity after 28 days at the level of 92.0%, and after 90 days at the level of 108.8% [15].

The observation of the morphology of the ash particles showed that they have regular shapes, mostly spherical [16]. From the point of view of the geopolymerization process, the presence of particles of such shapes in the fly ash is advantageous—it improves the rheological properties of the mixture, in particular improves its workability and reduces the need for the addition of liquid substances, which in turn has a positive effect on the seasoning and ensures obtaining geopolymeric materials with good properties mechanical [16].

Three types of fibers with different properties were selected for the study: coir, glass, and carbon fibers. Purchased coir fiber (Cocas nucifera), extracted from coconut shell, was used for the research. The coir is obtained from the outer layer of the coconut that surrounds the seed, which houses the actual palm seed [17,18]. This fiber is usually the waste that has traditionally been used to make mats, carpets, ropes, etc. However, only a small part of the produced fiber was used for these purposes. Currently, thanks to its property, it has been used in many industries, including construction industry [19,20]. The basic component of the fiber is lignin and it constitutes from 41 to 45% by weight of the fibers, the other ingredients are: cellulose—from 36 to 43% and hemicellulose about 0.2% [19,20,21]. Due to the high content of lignin, the fiber is durable and its stiffness and hardness increase with its content. It is also worth noting that coconut fiber has the highest natural fiber, the so-called microfiber angle, thanks to which it achieves a much higher elongation at break than other typical natural fibers [19,20]. It is also resistant to sea water and most bacteria and fungi. The basic physical and mechanical properties of coir are presented in Table 4 [19]. The length of purchased coconut fibers is from 10 to 33 cm, diameter 0.05–0.45 mm. The added fibers were crushed into pieces about 5 mm long (Figure 1a).

Due to its availability E-glass fiber chopped was used for the tests. Type E glass fiber is based on boron-aluminum-silicon glass, also known as alkali-free glass. Compared to other types, it is the cheapest type of fiberglass and the most readily available on the market. The strength properties of glass fiber are the better, the smaller its cross-section is. This fiber has high tensile strength and relatively low density. Glass fibers are sensitive to water and moisture, because it washes away alkali metal salts, creating gaps in the surface layers, therefore it is necessary to cover them with protective layers [22]. Glass fiber of type E has a lower value of Young’s modulus than e.g., fibers of S or R type, it results in a reduced stiffness of composites, which is important especially in the production of composites with a polymer matrix, and less important for composites based on cements or geopolymers. Its advantage, compared to carbon fiber, is its greater ability to absorb energy [22]. The basic physical and mechanical properties of glass fiber are presented in Table 4 [22,23]. The length of the glass fibers purchased is approximately 3 mm, diameter 10 μm (Figure 1b).

The increasing use of carbon fibers is related to their numerous advantages, they are: low density, high tensile strength and high Young’s modulus, high fatigue strength and creep strength, abrasion resistance, non-melting properties, high chemical resistance, high dimensional stability, good electrical conductivity, low friction coefficient, vibration damping capacity and low absorption of X-rays [24,25]. The disadvantages of carbon fibers include the tendency to oxidize. However, it should be noted that the oxidation of carbon fiber is catalyzed by the alkaline environment [25]. This property makes the geopolymer matrix, having an alkaline reaction, a promising possibility for the production of composites, which can also work at high temperatures. The basic physical and mechanical properties of carbon fiber are presented in Table 4 [23,24]. The purchased carbon fibers are approximately 5 mm long and approximately 8 μm in diameter (Figure 1c).

### 2.2. Sample Preparation

The samples were made on the basis of fly ash with three types of fibers with their different percentages and for the control sample—without the addition of fibers. Fibers were added in an amount of about 1, 2, and 5% by weight to the fly ash used as the geopolymer matrix base, except the carbon fibers, where applied only 1 and 2% by weight. The research on the 5% addition of carbon fiber was abandoned due to problems related to the sample production process—too high density of the mixture, which did not allow for proper mixing of the ingredients, ensuring the proper distribution of fibers in the matrix of the composite.

The first step was to mix the fly ash with the fibers. The test samples were labeled depending on the type and amount of fibers—Table 5.

A solution of sodium hydroxide (NaOH) with a molar concentration of 8M containing the addition of sodium water glass was added to the mixture previously prepared (fly ash and reinforcement). It was decided to work with 8M solution due to the possibility of the potential reaction of the alkaline solution with natural fibers—coconut and glass fiber. The sodium base solution was obtained from flakes of technical sodium hydroxide combined with an aqueous solution of sodium silicate R-145 (ratio of sodium base to water glass: 1:2.5). Tap water was used to prepare the solution. The resulting solution was mixed thoroughly and allowed to equilibrate to a constant concentration and temperature before combining with the solids of the mixture.

The obtained mass was mixed for 15 min in a slow-speed mixer, then it was transferred to the mold and compacted on a vibrating table. Then the molds were cured for 24 h at the temperature of 75 °C. After this time, the samples were cooled to ambient temperature, demolded, and stored for 6, 13 and 27 days (7, 14, or 28 days from production depending on the planned test time). Seasoning took place under laboratory conditions.

### 2.3. Methods

Compressive strength tests were carried out according to the methodology described in the standard EN 12390-3: Testing hardened concrete. Compressive strength of test specimens’. The tests were carried out on the Matest 3000 kN (Matest, Treviolo, Italy) device on cubic samples of 50 mm × 50 mm × 50 mm stored at ambient temperature for 7, 14, and 28 days. Each composition was tested on 15 samples. In accordance with the requirements of the standard, the samples were cleaned before starting the test (dirt and loose particles were removed so that they did not come into contact with the pressure plates). Due to the selected method of seasoning, it was not necessary to remove excess moisture from the samples. The specimen was positioned in the center of the lower platen so that the applied load was perpendicular to the specimen forming direction (top mold surface). A load speed of 0.5 MPa/s was assumed. The take-off force of the load was 0.5 kN, the load was continuously increased until the maximum value was obtained. The device security was set at 30%.

Flexural strength tests were carried out according to the methodology described in the standard EN 12390-5 (‘Testing hardened concrete. Flexural strength of test specimens’). Bending strength tests were carried out on the Instron type 4465 universal testing machine (Instron Companies, Norwood, MA, USA) and on the Matest 3000 kN (Matest, Treviolo, Italy) device on prismatic samples of 50 mm × 50 mm × 200 mm, stored at ambient temperature for 7, 14 and 28 days. Tests of each composite were carried out on 5 samples. In accordance with the requirements of the standard, before starting the test, the samples were cleaned so that the rollers were in direct contact with the sample. The samples were centered and the longitudinal axis of each sample was at right angles to the longitudinal axis of the upper and lower rollers. The adopted load direction was perpendicular to the sample forming direction. A loading speed of 0.05 MPa/s was assumed. The take-off force of the load was 0.001 kN, the load was continuously increased until the maximum value was obtained. A centric method of loading the samples was adopted (the so-called three-point bending). The used spacing of the lower rollers l = 150 mm. The device security was set at 30%.

The morphology of the samples was analyzed on the material remaining after the strength tests, both after the compressive strength and bending strength tests. A JEOL JSM-820 scanning electron microscope (IXR Inc., Austin, TX, USA) was used for the research. Before the test, small amounts of the materials were dried to constant weight and then placed on a carbon bed to drain the sample charge. The materials were sprayed with a thin layer of gold using a JEOL JEE-4X sputtering machine. The observations were made at various magnifications (50–2000×).

## 3. Results

### 3.1. Compressive Strength

Compressive strength values are summarized in Table 6.

For compositions with coconut fibers, the results show the increase in compressive strength for each type of sample over time. Best results are achieved after 28 days. Significant due to the properties of the materials also seems to be the increase in the repetitiveness of the results over time (reduction of the standard deviation after 28 days, compared to the results obtained after 7 and 14 days). This makes the material more predictable in the intended applications. The highest values were recorded for the addition of 2% coconut fibers after 28 days. Compared to the matrix material, there was an improvement of 43.3%. For the values of 1 and 5%, a significant increase in the value was also noted, by 27.7% and 31.5%, respectively. From the point of view of compressive strength, the 2% addition of coconut fiber seems to be optimal.

The tests carried out on composites with coir, glass, and carbon fibers show an increase in compressive strength for each type of samples with time. Best results are achieved after 28 days. The exceptions are samples with 1% addition of glass fiber, where the strength practically does not change with time, and any differences are smaller than statistical errors. The best results were achieved after 28 days for the 5% addition of glass fibers. With regard to the matrix material, there was an improvement of 47.9%. A slightly lower value was obtained for the composite with 2% glass fiber addition—there was an improvement of 47.10%. For these two additives, it is worth considering the possibility of investigating intermediate values, i.e., additives in amounts of 3 and 4%. When choosing between the addition of 2 and 5%, the price of the fiber will be important. It is higher than the price of the matrix, which means that the addition of 2% glass fiber, providing similar properties, will be more likely to be used from an economic point of view. The improvement in compressive strength also occurred with the 1% additive by weight of glass fiber, but was lower than with the 2 and 5% additives. With regard to the matrix material, there was an improvement of 25%. It is also worth paying attention to the fact that the addition of fibers can accelerate the achievement of high early strength for materials, it is especially visible for the 1 and 5% addition of glass fiber. This property is desirable in many construction applications where commissioning time is important, including repairs, e.g., high speed ways.

The tests carried out on composites with carbon fibers show the increase in compressive strength for each type of sample with time. Best results are achieved after 28 days. The best results were achieved with the 2% carbon fiber addition. Compared to the matrix material, there was an improvement of 56.5%, slightly lower values were obtained for the addition of 1%, in this case there was an improvement of 51.3%. One should also pay attention to the high strength properties of the composites after 7 days.

Significant improvement was obtained for all types of fibers regardless of their addition. The highest values were recorded for samples with carbon fiber with the addition of 2% and 1%, respectively. The addition of natural—coir fibers in the amount of 2% strengthened the geopolymer matrix in a similar way as the addition of glass fibers in the amount of 2 and 5%. The obtained results in the field of compressive strength tests show a positive effect of the addition of fibers and a significant increase in strength.

### 3.2. Bending Strenght

The bending strength values are summarized in Table 7.

The tests carried out on composites with coir fibers show that the improvement in bending strength was achieved only for the addition of 2% coconut fibers. Compared to the matrix material, there was an improvement of 5.4%. In the case of admixture: 1 and 5%, the bending strength deteriorated compared to the geopolymeric matrix material. As in the case of compressive strength, also in terms of bending strength, we deal with the stabilization of material properties over time. After 28 days, there is an increase in the reproducibility of the results over time (reduction of the standard deviation). It is also worth noting that the results after 28 days for selected samples, i.e., the matrix material and the composite with 2% fiber addition, deteriorated, but this change is small and is within the statistical error limit.

For glass fiber compositions, the results show the increase in bending strength for each type of specimen over time. Best results are achieved after 28 days. The improvement in bending strength occurred only for the 1% addition of coconut fibers. Compared to the matrix material, there was an improvement of 4.5%. For the addition of 1%, the same value was achieved with the geopolymer matrix material, and for the addition of 5%, the bending strength deteriorated. It is worth paying attention to the even values for composites with different amounts of glass fibers after 28 days, the differences are within the statistical error.

For carbon fiber compositions, the results show the increase in bending strength for each type of specimen over time. Best results are achieved after 28 days. For carbon fibers, there has been a significant improvement in bending strength. In the case of the addition of 1%, the bending strength increased by 62.4% in relation to the geopolymeric matrix material, and for the addition of 2% of glass fibers, by as much as 115.6%. It is also worth paying attention to the high values of flexural strength for these composites, achieved after just 7 days.

In terms of flexural strength, a significant improvement in the properties of the composites is seen with the addition of 1 and 2% carbon fibers. A slight increase in bending strength is also visible for 1% addition of glass fiber and 2% of coconut fibers. In the remaining cases, the composites obtained the bending strength values the same or lower than the geopolymer matrix material. No change in the nature of fractures from brittle to ductile was observed during the research. The behavior of composites with the addition of fibers is similar to the matrix material itself, where brittle fracture occurs.

The obtained results in the field of bending strength show a positive effect of the addition of carbon fibers in the amount of 1 and 2%, 1% of glass fiber and 2% of coconut fibers on the bending strength. In the case of other additives, this value does not change or it deteriorates.

### 3.3. Microstructure Investigation

The SEM analysis, carried out at high magnification, showed the structure of the coir fiber, which has a structure typical of most natural fibers, i.e., it has a rough structure (Figure 2a). The observations also showed that the matrix material is not always fully coherent with the fibers. In Figure 2b, there are visible discontinuities between the fiber and the geopolymer matrix that can adversely affect the mechanical properties of the composites.

Observation of the microstructure revealed potential problems that may exist in the composite: lack of matrix-fiber coherence. In this case, it is possible to reflect on the merits of pretreating the fiber by soaking it in water or low-molar alkali, as appropriate, before applying it to the composite, which may improve the continuity of the material. The other research with applying natural fibers in geopolymers confirm that cohesion between fiber and matrix could be improved by using proper pre-treatment [25,26].

SEM analysis, carried out at high magnification, showed the surface structure of the glass fiber, having a smooth surface and constant dimensions (Figure 3a). Glass fiber is a chemical fiber, characterized by repeatability, in contrast to natural fibers, i.e., coir fiber, whose individual fibers can significantly differ in dimensions and structure. In the composite, SEM observations showed no problems with matrix-fiber coherence. The observations carried out at lower magnifications showed agglomeration of fibers appearing in the composite (Figure 3b).

Carbon fiber, similar like glass fiber, has a smooth surface and constant dimensions (Figure 4a). It is characterized by dimensional repeatability. SEM observations of composites with carbon fibers revealed no problems with the matrix-fiber coherence (Figure 4a). No gaps or cracks were observed between the fiber-matrix system, even in high magnification observations.

In composites with carbon fibers, as in the case of composites with glass fibers, agglomerations of fibers were observed (Figure 4a). As in the case of glass fibers, this phenomenon may be related to the technology of fiber production. The occurrence of fiber agglomeration may be associated with many factors such as too short mixing time, inappropriate spindle for mixing for particular types of fibers or using too long fibers. Due to the manufacturing process used, chemical fibers (glass and carbon) very often have the form of several strands of fibers joined together. In this case, it is advisable to use a longer mixing time than in the case of natural fibers or other fibers produced using different technologies, i.e., steel fibers. In composites with the addition of carbon fiber, it was possible to observe phenomena related to the propagation of cracks in the material (Figure 4b). Fibers added to the geopolymer matrix prevent their development, which delays the failure mechanism of the material.

## 4. Discussion

The results of the tests of mechanical properties after 28 days for composites are summarized in Table 8.

The research showed an increase in compressive strength for composites with fibers—improvement in properties between 25.0% and 56.5% depending on the type and amount of added fiber. For the bending strength, a clear increase in the strength value is visible for composites with the share of carbon fibers in the amount of 1 and 2% (62.4% and 115.6%). There was also a slight increase in bending strength for 1% addition of glass fiber (4.5%) and 2% addition of coconut fibers (5.4%). For a 2% addition of glass fibers, the bending strength value did not change compared to the value obtained for the matrix material. For the remaining fiber additives, i.e., 5% of glass fibers and 1 and 5% of coconut fibers, the bending strength values deteriorated—a decrease between 1.9% and 27.7%.

Comparing the obtained results with other studies carried out with the use of the same fibers, it is worth noting that in the case of coir fibers, the obtained results are partially consistent with previous research [17,18,27,28]. According to the literature review, the most advantageous additive due to its mechanical properties is 0.5–1% by weight. Our own research did not confirm this proportion. The best results were obtained for 2% by weight of the fiber, for lower values, worse results for both bending strength and compressive strength were obtained. However, studies on the content of more than 1% by weight of coir fiber have not been described in the available literature so far [17,18,27,28].

Previous research work has shown that the addition of coir fibers can improve the compressive strength between 8.2% and 26.5%. In the conducted research, it was possible to obtain an improvement in the properties of the matrix material between 27.7% and 43.3%, which is even higher than the result described in the literature. However, other fly-ash was used in reference studies as a raw material and it could also have an influence on achieved results as well as other parameters during the preparation process. It is worth noting that the obtained value for the composite reinforced by coir fiber was comparable with other values whose compressive strengths were between 5.7 MPa and 80.7 MPa. In the case of extreme values, it should be noted that for the obtained compressive strength value of 5.7 MPa [28], the main aim of the study was not to obtain geopolymers of high strength, but the possibility of waste disposal in geopolymeric materials. In the research conducted obtaining high values of the compressive strength of the matrix was not the aim of the research, therefore it was decided to use a matrix based on a geopolymer without the addition of aggregates, which allowed for the elimination of an additional factor that could potentially affect the results of the research. Additionally, it was decided to use a solution with a low molar concentration—8M, to avoid problems with dissolving the fibers [29]. It should be noted that the source works indicate that the compressive strength of geopolymers increases with the increase of the molar concentration of the alkaline solution and the temperature used in the geopolymer production process, although it should be noted that after exceeding certain concentrations, this value may decrease [30], and the process of geopolymerization in too high concentrations may not proceed properly, and as a result, an alkaline activated material with a two-dimensional lattice will be obtained [31]. Bending strength tests presented in the literature show different behavior of composites for this property. The admixtures of fibers does not always improve bending strength of composites. In cases where it increased, the improvement was small by 2.7% compared to the matrix material [17]. Own research confirmed this behavior of composites with coir fibers.

In the case of glass fibers, the optimal values of the additives by weight for glass fibers most often quoted are between 0.5 and 1%. The conducted research confirms that the addition of 1% has a positive effect on both the compressive strength and bending strength of composites. In the case of higher additives, the compressive strength improved, but the bending strength deteriorated. Earlier research studies indicate that in the case of compressive strength, the improvement in mechanical properties was between 6.4% and 16.6% [30,32,33,34]. The results of own research showed an improvement in this property and between 25 and 47.8%. Therefore, it was higher than in the literature. In the case of bending strength, the literature sources provide the obtained values of improvement of this property between 16.4 and 33.8% [30,32,33,34]. In the conducted tests, it was possible to obtain only a slight improvement in bending strength—4.5%. It must be stressed that the mechanical properties for the geopolymer composite reinforced by glass fiber will be strongly depended on type of fiber (it must be alkali resistant, the best is type AR) and molar solution of used alkali (high molar ratio easier destroyed the glass fibers).

The results obtained for composites with carbon fiber are consistent with the source data, while earlier studies used very different carbon fiber additives from 1% to even 10% in the case of using microfibers [35,36]. In the conducted research, it was possible to add carbon fibers in the amount of 1 and 2%. The tests with 5% carbon fibers were abandoned due to problems with the workability of the mixture. The obtained values of the improvement of mechanical properties are similar, both in terms of compressive strength and bending strength, to the data known from the literature [37,38].

The values of the obtained properties and mechanical properties can be compared to traditional building materials. The value obtained for the geopolymer matrix can be compared to the properties of the popular concretes known as C16/20 (according to: PN-EN 12390-3: 2011). According to the requirements of the C16/20 standard, for cubic cubes, the results of the compressive strength of 20 MPa. This type of concrete should have a bending strength above 4 MPa (PN-EN 12390-5: 2011). Concrete C16/20 is currently one of the most commonly used type of concrete, it can be used practically at every stage of the building a house, from pouring foundation footings, foundation plates, or the foundations themselves, as well as for various types of foundations, walls, columns, lintels, terraces, stairs, ceilings, etc. It is used in both internal and external applications, including for pouring and embedding fence posts and for making small concrete elements and small architecture [37,38]. Taking into consideration mechanical properties for all these applications, the geopolymeric material can be also used. The main problems in the application of geopolymeric composites in practice is the lack of appropriate standards for these materials [39]. Additional problem can be a fluctuation of the price raw materials, such as sodium silicate and sodium hydroxide [40]. When mentioning the applications of geopolymeric materials, it should be noted that they can also be strengthened in a traditional way, just like cement. However, in contrast, steel core does not cause corrosion in geopolymer composites [2,4].

Additionally, the addition of fibers to all composites increased their compressive strength to higher classes, i.e., C20/25 and 30 MPa C25/30 (according to: PN-EN 12390-3: 2011). It is especially worth paying attention to composites containing 2% coconut fiber, 2 and 5% glass fibers, and 1 and 2% carbon fibers, which meet the requirements for class C25/30, i.e., compressive strength of 30 MPa and bending strength of 6 MPa. These types of concretes are used in similar areas as before described C16/20 concretes, and are also used for: bicycle paths, sidewalks, forest roads, and elements of sports facilities. It is also worth noting that carbon fiber composites meet the requirements of the so-called architectural concretes, i.e., in bending of 8 MPa, which allows them to be used for complex spatial elements [39].

Currently, an important direction of development is the creation of new materials with the natural environment in mind. There is a significant market demand for ecological materials, and in particular for materials based on renewable raw materials. Modern composites, based on geopolymers, allow to reduce the emission of substances harmful to the environment and at the same time save natural resources. Geopolymer composites, in particular with reinforcements made of natural fibers, are part of the sustainable development policy, which is currently a guideline for the creation of legal standards in many countries in Europe and in the world [1,14,41].

## 5. Conclusions

The achieved results show influence of all types of fibers on mechanical properties of the geopolymer composites. The increase in compressive strength for composites with fibers was between 25.0% and 56.5% depending on the type and amount of fiber added. For bending strength, a clear increase in the strength value is visible for composites with 1 and 2% carbon fibers (62.4% and 115.6%). A slight improvement of bending strength was obtained also for 1% addition of glass fiber (4.5%) and 2% addition of coconut fibers (5.4%). For the 2% addition of glass fibers, the flexural strength value did not change compared to the plain matrix material. For other compositions, such as 5% glass fiber as well as 1 and 5% coconut fibers, the bending strength values deteriorated. The observed behavior is coherent with results presented in the literature.

New composites based on geopolymer, reinforced with coconut, glass and carbon fibers, were obtained, and the possibilities of their use as construction materials, in particular as materials for use in construction, were determined. The limitations related to the use of new materials were also investigated. The possibility of using particular types of fibers as reinforcement for selected composites with a previously prepared geopolymer matrix was assessed, depending on their type and quantity as well as the seasoning time of the samples. The structure and mechanical properties of the composites based on geopolymers were analyzed and their behavior depending on the amount of reinforcement was determined. Then, on the basis of the obtained results, the possibilities of using the new composites as an environmentally friendly construction material were described. For this purpose, the analysis of conventional materials used so far for selected construction applications was performed and their properties compared with the obtained composites. The conducted analyses indicate that the selected composites based on a geopolymer reinforced with fibers has potential applications as a construction material or it could be mixed with fine or coarse aggregate and used as geopolymer concreate for the elements where higher mechanical properties are required.

## Figures and Tables

**Figure 1 materials-14-04599-f001:**
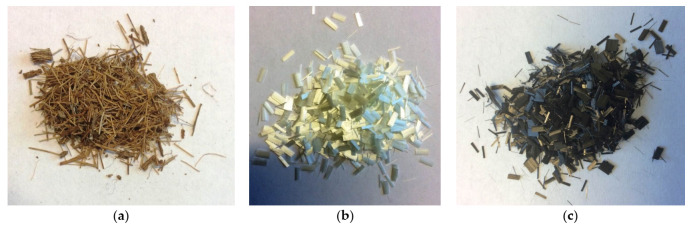
Short fibers (**a**) coir (after crushed), (**b**) glass fiber, and (**c**) carbon fiber.

**Figure 2 materials-14-04599-f002:**
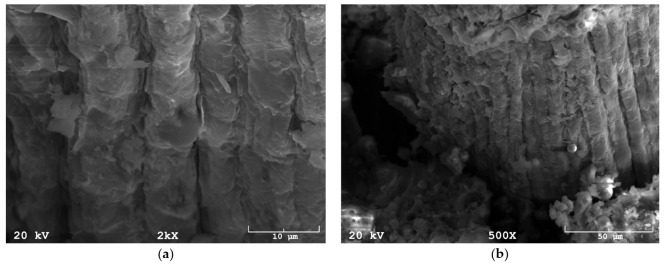
SEM image: (**a**) coir structure in a geopolymer matrix, (**b**) coherence of the fiber with the geopolymer matrix.

**Figure 3 materials-14-04599-f003:**
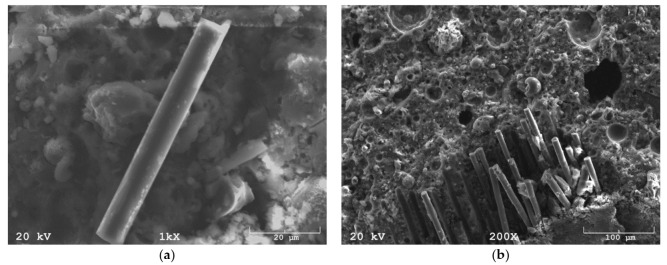
SEM image: (**a**) glass fiber in a geopolymer matrix, (**b**) fibers agglomeration.

**Figure 4 materials-14-04599-f004:**
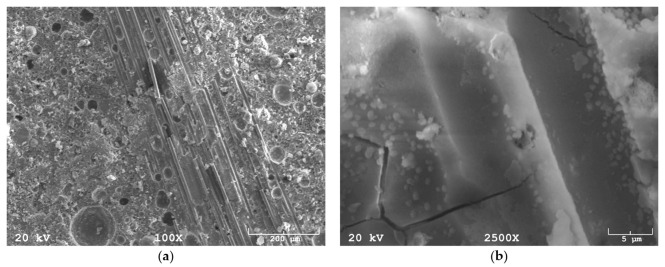
SEM image: (**a**) carbon fiber in a geopolymer matrix—agglomeration, (**b**) composite with the addition of carbon fibers, the photo shows the space left by the carbon fibers.

**Table 1 materials-14-04599-t001:** Nowadays commercial application of geopolymers.

No	Company	Product	Kind of Product
1	PCI BASF, Heidelberg Germany	PCI Geofug^®^	Geopolymer grount and binder
2	Milliken Infrastructure Solutions, LLC, Spartanburg, SC USA	GeoSpray™	Geopolymer mortar for pipe rehabilitation technology
3	GEOFIP, Kyiv Ukraine	GEOFIP™	Geopolymer paints, adhesives and coatings
4	ASK CHEMICALS, Hilden, Germany	INOTEC—Inorganic binder system	Geopolymer binder for foundries
5	Wagners, Queensland, Australia	Earth Friendly Concrete	Geopolymer binder system made from industrial waste by-products: blast furnace slag and fly ash
6	Pyromeral System, Barbery, France	Pyromeral Systems	Alumino-silicate-based geopolymeric composites (fire and heat resistant materials)
7	Skoberne, Pfungstadt, Germany	SKOBIFIX 30	Geopolymer foam dedicated for heating systems
8	Nu-core^®^, Canberra, Australia	Nu-Core^®^A2FR	Geopolymer fireproof composite panels
9	RENCA, Moscow, Russia	RENCA Geopolymer 3D ink	Geopolymer 3D printing mortar
10	INOMAT, Neunkirchen, Germany	Ino-Flamm^®^	Fire resistant geopolymer paint
11	Sinotec, Wiesbaden, Germany	Sinnocoat^®^ Geopolymer HT	Anti-corrosive geopolymer coating
12	GeoPol, Holubice, Czech Republic	GeoPol	Geopolymer sand binder for cores in foundries
13	Allied Foam Tech Corp., Montgomeryville, Pa, USA	GeoFoam	Geopolymer foam Cement for geotechnical applications
14	Vodnis Klo, Prage, Czech Republic	Desil Al	Binder systems to foundry industry
15	Wöllner GmbH, Ludwigshafen, Germany	GEOSIL^®^	Binders and hardeners for alkali-activated systems
16	RCPA, Beenleigh QLD, Australia	eCP	Geopolymer concrete pipes
17	Zeobond, Melbourne, Australia	E-Crete™	Precast geopolymer concrete
18	Watershed Materials, Napa, CA, USA	Watershed Block™	Geopolymer blocks
19	BowersIndustrial, West Jordan, UT, USA	GEOPOLYMER-A400 GEOPOLYMER-A600	Geopolymer concrete material designed for resistance to acetic acid
20	Aquaminerals, Paltamo, Finland	Ammoniacal nitrogen	Adsorbent based on geopolymer technology
21	Betolar, Kannonkoski, Finland	Betolar	Cement-free construction material
22	John Wood Group PLC, Scotland, United Kingdom	SIAL^®^	Encapsulating radioactive waste streams
23	Lucideon Ltd., Staffordshire United Kingdom	iCRT	Encapsulating hazardous wastes
24	Rocla, Sydney, Australia	---	Crypt components
25	Murray & Roberts Cementation Co. Ltd., Bedfordview, South Africa	HVPFAC	Geopolymer concrete
26	URETEK, Pirkkala, Finland	---	Different solutions, i.a. pillars, geopolymer injection

**Table 2 materials-14-04599-t002:** Oxide composition of fly ash.

	LOI	SiO_2_	Al_2_O_3_	Fe_2_O_3_	CaO	MgO	Na_2_O	K_2_O	SO_3_	TiO_2_	P_2_O_5_	BaO
[%]	2.44	55.89	23.49	5.92	2.72	2.61	0.59	3.55	0.16	1.09	0.82	0.20

**Table 3 materials-14-04599-t003:** Fine particle fraction.

Particle Fraction	>160 μm	100–160 μm	71–100 μm	63–71 μm	56–63 μm	<56 μm
Amount	0.3%	3.2%	11.9%	9.9%	15.4%	59.3%

**Table 4 materials-14-04599-t004:** Selected physical and mechanical properties of fibers [17,18,19,20,21,22,23,24].

	Coir	Glass Fiber	Carbon Fiber
Density	1.25–1.5 g/m^3^	2.54 g/m^3^	1.6–2.0 g/m^3^
Young Modulus	4–6 GPa	60–70 GPa	230 GPa
Tensile strength	95–175 MPa	1350–3500 MPa	2800–5000 MPa
Elongation	17–51.4%	1.5–3.5%	1–1.5%

**Table 5 materials-14-04599-t005:** Composition of the samples.

Mark	Sample Composition
0	Reference sample (pure matrix)
K 1%	Geopolymer with 1% coir
E 1%	Geopolymer with 1% glass fiber
C 1%	Geopolymer with 1% carbon fiber
K 2%	Geopolymer with 2% coir
E 2%	Geopolymer with 2% glass fiber
C 2%	Geopolymer with 2% carbon fiber
K 5%	Geopolymer with 5% coir
E 5%	Geopolymer with 5% glass fiber

**Table 6 materials-14-04599-t006:** Compressive strength—tested geopolymer composites after 7, 14, and 28 days.

	0	K 1%	E 1%	C 1%	K 2%	E 2%	C 2%	K 5%	E 5%
7 days									
Compressive strength [MPa]	17.7	22.7	29.2	29.7	21.8	21.2	29.4	20.2	28.8
Standard deviation	5.8	8.0	3.1	5.3	3.9	3.5	2.7	4.2	7.5
14 days									
Compressive strength [MPa]	22.5	29.4	29.0	29.8	32.1	27.4	33.6	22.8	30.5
Standard deviation	2.0	9.4	7.2	4.5	6.1	6.2	6.1	3.4	6.1
28 days									
Compressive strength [MPa]	23.3	29.8	29.1	35.3	33.4	34.3	36.5	30.6	34.5
Standard deviation	4.4	6.1	2.6	6.9	5.1	2.8	4.8	3.9	3.0

**Table 7 materials-14-04599-t007:** Bending strength—tested geopolymer composites after 7, 14, and 28 days.

	0	K 1%	E 1%	C 1%	K 2%	E 2%	C 2%	K 5%	E 5%
7 days									
Bending strength [MPa]	5.2	4.8	6.2	8.9	6.1	5.5	9.6	4.4	5.7
Standard deviation	0.5	0.4	0.5	2.2	1.2	0.7	0.9	0.8	1.0
14 days									
Bending strength [MPa]	5.0	5.3	6.4	9.4	6.6	5.6	11.4	4.3	5.6
Standard deviation	0.7	1.0	0.6	3.3	0.2	0.6	3.4	0.3	0.7
28 days									
Bending strength [MPa]	6.2	5.3	6.4	10.0	6.5	6.2	13.3	4.4	6.0
Standard deviation	0.6	0.4	1.0	1.0	0.3	0.5	3.9	0.7	0.7

**Table 8 materials-14-04599-t008:** Summary of composite test results—compressive and bending strength after 28 days.

Sample	Compressive Strength	% Improvement (Compare to Reference Samples)	Bending Strength	% Improvement/Deterioration (Compare to Reference Samples)
0	23.3 ± 4.4	---	6.2 ± 0.6	---
K 1%	29.8 ± 6.1	27.7	5.3 ± 0.4	−13.2
E 1%	29.1 ± 2.6	25.0	6.5 ± 1.0	4.5
C 1%	35.3 ± 6.9	51.2	10.0 ± 1.0	62.4
K 2%	33.4 ± 5.1	43.3	6.5 ± 0.3	5.4
E 2%	34.3 ± 2.8	47.1	6.2 ± 0.5	0.0
C 2%	36.5 ± 4.8	56.5	13.3 ± 3.9	115.6
K 5%	30.6 ± 3.9	31.5	4.4 ± 0.7	−27.7
E 5%	34.5 ± 3.0	47.8	6.0 ± 0.7	−1.9

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
