# Peer review of "The Influence of Short Coir, Glass and Carbon Fibers on the Properties of Composites with Geopolymer Matrix"

_materials, 2021, doi:10.3390/ma14164599_

Round 1

Reviewer 1 Report

The authors investigated the influence of short coir, glass and carbon fibres on the strength of the geopolymer composites. The topic is interesting and fits well with the scope of the journal, but the article should be improved  before publishing it. Here are some comments that could help to improve the quality of this work.

  1. Line 9, Abstract: “The aim of the article is to analyse the influence of short coir, glass and carbon fibre admixture on the mechanical properties of fly ash-based geopolymer, especially flexural and compressive strength.”
  • Only flexural and compressive strengths were investigated. You should replace “especially” by another correct word.
  1. Line 89: “The length of purchased coconut fibers is from 10 to 33 cm, diameter 0.05 - 89 0.45 mm.”
  • Based on Figure 2, the diameters of coconut fibers are greater than 100 µm. Each fiber is composed of fibrils (microfibers) bundle, where the diameter of each fibril is around 5 µm.
  1. Table 3
  • Spelling error “Bending strengtht”
  • Tensile strength is usually used instead of “bending strength” as a major characteristic of fibers, please check if this characteristic - as mentioned in the table- is correct.
  • If you did not carry out these results, then you should mention the sources at the end of the table title.
  1. 3 Methods
  • Write the model and the manufacturer country of each instrument.
  1. Line 175: “A JEOL 172 JSM-820 scanning microscope ….”
  • Is this Scan electron microscope?
  1. Line 264: “The breakthrough observations also showed that the matrix material is not 264 always fully coherent with the fibers.”
  • The breakthrough observations??; It is expected result because of the hydrophobic characteristics of most of natural fibers.
  1. Line 270: “In this case, it is possible to reflect on the merits of pretreating the fiber by soaking it in water or low-molar alkali, as appropriate, before  applying it to the composite, which may improve the continuity of the material.”
  • There are many studies about improving the adhesion characteristics of natural fibers by immersion in alkaline solution to remove the lignin. See https://doi.org/10.1016/j.clay.2017.03.030  https://doi.org/10.3389/fbuil.2021.631307
  1. Line 293: “In this case, it is advisable to use a longer mixing time than in the case of natural fibers or other fibers produced using different technologies, i.e. steel fibers.”
  • Mixing for long time could increase the fiber agglomeration. Usually, you need to use appropriate spindle for mixing or to introduce shorter fibers.
  1. Line 327: “the main aim of the study was not to obtain geopolymers of high strength, but the possibility of waste disposal in geopolymeric materials.”
  • I am not sure if the natural fibers are considered as waste.
  • This “new” aim was not mentioned in the introduction, see lines 45-51.
  1. Line 342: “It does not always improve.”
  • Please, write complete and clear sentence
  1. Lines 345-355: “In the case of glass fibers, …..”
  • Check in the literature the effect of alkaline medium on the strength of glass fiber.
  1. Line 396: Conclusions
  • Summarize your results

Author Response

Thank you for valuable suggestions, the following changes was implemented into manuscript:

  • Line 9, Abstract: “especially” was replaced by such as.
  1. Line 89: 100 µm =  0.1 mm à “diameter 0.05 - 0.45 mm.” =  40 – 450 µm. We clarify description under the figure.
  1. Table 3
  • Spelling error “Bending strengtht” was corrected on “Tensile strength”.
  • The sources was added.
  1. 3 Methods
  • The model and the manufacturer country of each instrument is supplemented.
  • Line 175: Is this Scan electron microscope? Yes, it is. Information is added.
  1. Line 264: The sentence has been changed.
  1. Line 270: Thank you for valuable suggestion. Both of them was used in article and added to references.
  1. Line 293: The sentence was reformulate and includes reviewer suggestions.
  1. Line 327: correction was made in the introduction part.
  • The information that the coir is a waste product from the coconut industry was added in introduction part.
  • The aim was clarified in the introduction part.
  1. Line 342: The sentence was improved.
  1. Lines 345-355: The information about the effect of alkaline medium on the strength of glass fiber and geopolymer composites is added.
  1. Line 396: The summaraizing of the results is added.

Reviewer 2 Report

The manuscript concerns the use of different types of fibres (coir, glass and carbon fibres) in fly ash geopolymer matrix at different percentages. The topic in interesting but the paper needs to be re-written in some parts.

They should highlight what is the aim of this study and its novelty. Furthermore, it is not clear the application of their materials. They made a comparison with geopolymer concretes while they proposed a geopolymer paste since they chose do not include aggregates in their mixtures.

1) Introduction: the section is too short and the literature review is very poor.  An extensive description about advantages and disadvantages and applications of geopolymers is necessary to introduce the topic analyzed. See for example:

Geopolymers and Fiber-Reinforced Concrete Composites in Civil Engineering. Polymers13(13), 2099, 2021.

An Application Review of Fiber-Reinforced Geopolymer Composite. Fibers9(4), 23, 2021.

Eco-efficient industrial waste recycling for the manufacturing of fibre reinforced innovative geopolymer mortars: Integrated waste management and green product development through LCA, 2021.

2) Line 321: The authors state that they obtained an improvement of the properties between 27.7% and 43.3% which are higher than the results found in literature.

“The Previous research work has shown that the addition of coir fibers can improve the compressive strength between 8.2% and 26.5%. In the conducted research, it was possible to obtain an improvement in the properties of the matrix material between 27.7% and 43.3%, which is even higher than the result described in the literature”

In order to make a comparison with other studies you should use a mix design very similar to the other research studies. What are the studies that you used as reference? Could you explain what are the differences from the other studies cited in literature? What are the components that differ from the other studies (if any)? How can you explain a so high improvement of the properties?

3) Line 329: The authors states that they didn’t use aggregates in their matrix so it can be defined as a geopolymer fiber-reinforced paste.

“In the research conducted as part of the dissertation, obtaining high values of the compressive strength of the matrix was not the aim of the research, therefore it was decided to use a matrix based on a geopolymer without the addition of aggregates, which allowed for the elimination of an additional factor that could potentially affect the results of the research.”

Why did the authors not add the aggregates in the matrix if they report in several parts of the manuscript that the final application of these kind of composites is the use as a building material?

4) Line 363: The authors state that the materials they prepared are comparable to the traditional building materials (they mention concrete) because of the values recorded for the mechanical properties.

“The values of the obtained properties and mechanical properties can be compared to traditional building materials. The value obtained for the geopolymer matrix can be compared to the properties of the most popular concretes known as C16/20 (according to: PN- 365 EN 12390-3: 2011).”

Concrete is a building material prepared with cement, aggregates and water. In your paper it seems that you didn’t prepare a geopolymer concrete but a geopolymer paste.

How can you compare your geopolymer paste to concrete? The two materials are not comparable because they are not similar.

5) The authors state that concrete C16/20 is the most commonly used type of concrete, with the most universal applications.

“Concrete C16/20 is currently the most commonly used type of concrete, with the most universal applications. It has found wide construction applications and can be used practically at every stage of the building a house, from pouring foundation footings, foundation plates, or the foundations themselves, as well as for various types of foundations, walls, columns, lintels, terraces, stairs, ceilings, etc. It is used in both internal and external applications, including for pouring and embedding fence posts and for making small concrete elements and small architecture”.

Could you please add some references to this statement? For civil applications, C25/30 concrete is widely versatile and used in numerous commercial and domestic projects while C16/20 is mostly used for lightweight structures and other applications due to its lower resistance.

6) In the conclusions the authors state that geopolymers can substitute concrete.

“The conducted analyses indicate that the selected composites based on a geopolymer reinforced with fibers are a construction material that has better mechanical properties than traditionally used building materials, i.e. concrete”.

As reported in my comment 3) the geopolymers prepared are not comparable with concrete and to state that they have better mechanical properties the authors should conduct also durability tests.

Furthermore, geo-polymerization process is very sensitive.

Author Response

Thank you for valuable suggestions, the following changes was implemented into manuscript:

Comment without numeration: The aim of this study,  its novelty, application of the materials were clarify in introduction part.

1) Introduction: the applications of geopolymers were added – please see Table 1. The suggested literature is implemented into the reference.

2) Line 321: The additional comment was added.

3) Line 329: The composites were prepared without aggregates, as described in samples preparation process.

4) Line 363: The other authors research presented increasing the mechanical properties after fine aggregate addition – sand: reference: 10.

5) The part connected with C16/20 is reformulated. New references are added.

6) The conclusion part is reformulated.

Round 2

Reviewer 1 Report

The  authors have adequately addressed the  comments.

Author Response

Thank you.

Reviewer 2 Report

The authors improved the manuscript but some points are still to clarify. 

1) They should give an idea about the applications of their materials. 

2) The comparison with concrete can be only mentioned as values reached but the authors can not state that their materials can be used in substitution of concrete. 

3) The applications they mention like foundations, pillars,... are made of concrete reinforced with steel rebars. The fibres are used only to restore the damaged parts but not to realize structural parts. So the authors should give some applications of their materials. 

Author Response

Thank you for valuable comments. The changed parts are marked in red colour. The following changes are introduced in the text:

1) The additional comment about way of application are implemented in conclusion part. 

2) The changes in comparison with concrete are applied in lines: 394-398.

3) Additional explanation about reinforcement by steel is added in lines:  398-400.
